# Identifying underlying medical causes of pediatric obesity: Results of a systematic diagnostic approach in a pediatric obesity center

Lotte Kleinendorst[1,2,3°], Ozair Abawi[1,4°], Bibian van der Voorn[1,4,5], Mieke H. T. M. Jongejan[6], Annelies E. Brandsma[7], Jenny A. Visser[1,5], Elisabeth F. C. van Rossum[1,5], Bert van der Zwaag[8], Mariëlle Alders[2], Elles M. J. Boon[2,3], Mieke M. van Haelst[2,3‡], Erica L. T. van den Akker[2,3‡]*

1 Obesity Center CGG, Erasmus MC, University Medical Center Rotterdam, Rotterdam, The Netherlands, 2 Department of Clinical Genetics, Amsterdam UMC, University of Amsterdam, Amsterdam, The Netherlands, 3 Department of Clinical Genetics, Amsterdam UMC, Vrije Universiteit Amsterdam, Amsterdam, The Netherlands, 4 Division of Endocrinology, Department of Pediatrics, Erasmus MC-Sophia Children's Hospital, University Medical Center Rotterdam, Rotterdam, The Netherlands, 5 Division of Endocrinology, Department of Internal Medicine, Erasmus MC, University Medical Center Rotterdam, Rotterdam, The Netherlands, 6 Department of Pediatrics, Obesity Center CGG, Franciscus Gasthuis, Rotterdam, The Netherlands, 7 Department of Pediatrics, Obesity Center CGG, Maasstad Ziekenhuis, Rotterdam, The Netherlands, 8 Department of Genetics, University Medical Center Utrecht, Utrecht, The Netherlands

☯ These authors contributed equally to this work.
‡ These authors also contributed equally to this work.
* e.l.t.vandenakker@erasmusmc.nl

**Data Availability Statement:** All relevant data are within the manuscript and its Supporting Information files.

## Abstract

### Background

Underlying medical causes of obesity (endocrine disorders, genetic obesity disorders, cerebral or medication-induced obesities) are thought to be rare. Even in specialized pediatric endocrinology clinics, low diagnostic yield is reported, but evidence is limited. Identifying these causes is vital for patient-tailored treatment.

### Objectives

To present the results of a systematic diagnostic workup in children and adolescents referred to a specialized pediatric obesity center.

### Methods

This is a prospective observational study. Prevalence of underlying medical causes was determined after a multidisciplinary, systematic diagnostic workup including growth charts analysis, extensive biochemical and hormonal assessment and genetic testing in all patients.

**Funding:** The author(s) received no specific funding for this work.

**Competing interests:** The authors have declared that no competing interests exist.

## Results

The diagnostic workup was completed in n = 282 patients. Median age was 10.8 years (IQR 7.7–14.1); median BMI +3.7SDS (IQR +3.3-+4.3). In 54 (19%) patients, a singular underlying medical cause was identified: in 37 patients genetic obesity, in 8 patients cerebral and in 9 patients medication-induced obesities. In total, thirteen different genetic obesity disorders were diagnosed. Obesity onset <5 years (p = 0.04) and hyperphagia (p = 0.001) were indicators of underlying genetic causes, but only in patients without intellectual disability (ID). Patients with genetic obesity with ID more often had a history of neonatal feeding problems (p = 0.003) and short stature (p = 0.005). BMI-SDS was not higher in patients with genetic obesity disorders (p = 0.52). Patients with cerebral and medication-induced obesities had lower height-SDS than the rest of the cohort.

## Conclusions

To our knowledge, this is the first study to report the results of a systematic diagnostic workup aimed at identifying endocrine, genetic, cerebral or medication-induced causes of pediatric obesity. We found that a variety of singular underlying causes were identified in 19% of the patients with severe childhood obesity. Because of this heterogeneity, an extensive diagnostic approach is needed to establish the underlying medical causes and to facilitate disease-specific, patient-tailored treatment.

## Introduction

Obesity is a multifactorial disease that has become one of the greatest health challenges of our time. [1] The prevalence of severe obesity in children and adolescents (as defined by the World Health Organization [2] and the International Obesity Task Force [3]; IOTF) was recently shown to range from 1.7% to 6.3% in several countries. [4] Body mass index is strongly influenced by genetic susceptibility with an estimated heritability of 40–70%. [5, 6] Most children and adolescents with obesity do not have singular underlying medical disorders causing their obesity, such as endocrine disorders, genetic obesity disorders, cerebral or medication-related causes. [7] The pathophysiologic mechanisms of the underlying medical conditions causing obesity are widely varied, leading to the suggestion to talk about "different diseases causing obesity" or "obesities". [8] Establishing an underlying diagnosis can give insight into the clinical course of the obesity, and lead to tailored monitoring and treatment. [9] In addition, it ends the diagnostic odyssey and can reduce the stigma that patients are confronted with. [10, 11] Since pharmacological treatment for patients with genetic defects affecting the leptin-melanocortin pathway (the hypothalamic system that controls appetite and energy expenditure) [11] is currently being evaluated in clinical trials, identifying these diseases becomes even more relevant. [8, 12] It is difficult to assess which patients should be evaluated for underlying causes. The current international clinical practice guideline for the evaluation and treatment of pediatric patients with obesity was published in 2017 by the Endocrine Society (ES). [13] In this guideline, clinicians are guided through the diagnostic process. After medical history-taking and physical examination, specific additional diagnostic steps are suggested depending on the findings. In short, endocrine evaluation is recommended in patients with reduced growth velocity; evaluation of hypothalamic obesity in patients with central nervous system (CNS)

injury, and re-evaluation of drug choice in patients using antipsychotic drugs. In selected cases, genetic testing is recommended, e.g., in patients displaying extreme early-onset obesity (<5 years) and severe hyperphagia, which are considered cardinal features of genetic obesity disorders. The genetic tests mentioned in the guideline range from karyotyping to DNA diagnostics for deficiencies in the leptin-melanocortin pathway. As of yet, studies that systematically screen for the underlying medical causes mentioned in the ES guideline in children and adolescents with obesity have not been performed. Previous studies on genetic obesity disorders report an underlying causative genetic defect in 2–5% of non-consanguineous pediatric patients with severe obesity, [13–15] but prevalence of the other underlying medical causes of obesity has not been studied. Therefore, our primary aim was to analyze the results of a thorough diagnostic workup in a cohort of patients who had been referred to the pediatric division of a specialized tertiary obesity center. Our diagnostic approach included broad evaluation for each patient of all possible underlying medical causes of obesity as mentioned in the ES guideline: endocrine and genetic disorders, as well as cerebral injury and medication use. Moreover, we compared the detailed clinical phenotype of these patients to evaluate whether the patients with underlying medical causes of obesity can be distinguished from those without an underlying medical cause.

## Methods

For this analysis, medical data of children and adolescents aged 0–18 years visiting Obesity Center CGG (Dutch: *Centrum Gezond Gewicht*; English: *Centre for Healthy Weight*) were analyzed. Obesity Center CGG is a Dutch multidisciplinary referral center for obesity consisting of a collaboration between the departments of Pediatrics, Internal Medicine and Surgery of the academic hospital Erasmus MC and collaborating general hospitals Maasstad Ziekenhuis and Franciscus Gasthuis. In this prospective, observational study, informed consent was obtained at the initial visit according to Dutch law: written informed consent was obtained from parents and children >12 years; for children below age 12 years oral assent was additionally obtained. This also included separate consent forms for genetic testing. The study was approved by the medical ethics committee of the Erasmus MC (MEC-2012-257). Pediatric patients were referred to Obesity Center CGG for diagnostic evaluation (due to suspicion of underlying causes of obesity, severe obesity, or resistance to combined lifestyle intervention), personalized therapeutic advice, or participation in a combined lifestyle intervention (Fig 1). [16] All consecutive patients who provided written informed consent were included at the university medical center Erasmus MC-Sophia Children's Hospital from 2015 to August 2018. From 2016 to August 2018, the collaborating general hospital Maasstad Ziekenhuis also included patients with a suspicion of an underlying medical cause of obesity. Exclusion criteria for this study were inability or refusal to give informed consent, refusal to undergo genetic testing, or not completing the standardized diagnostic approach (Fig 1). A standardized diagnostic approach was applied for all patients (Fig 2), discussed below and in more detail in the S1 Appendix, aimed at identifying underlying endocrine, genetic, cerebral, and medication-induced main causes of obesity. At study entry, medical history-taking, physical examination and extensive assessment of growth charts were performed by a pediatric endocrinologist or pediatrician supervised by a pediatric endocrinologist. A few weeks after the initial visit, patients returned to the outpatient clinic where blood was drawn after an overnight fast for biochemical and hormonal evaluation, and genetic diagnostics. All patients and/or their parents were asked to fill out several questionnaires regarding physical activity, eating behavior, sleeping behavior, stress, and quality of life. Furthermore, all patient records were screened by a clinical geneticist. In case of high suspicion of genetic obesity or abnormal genetic test results, patients were seen

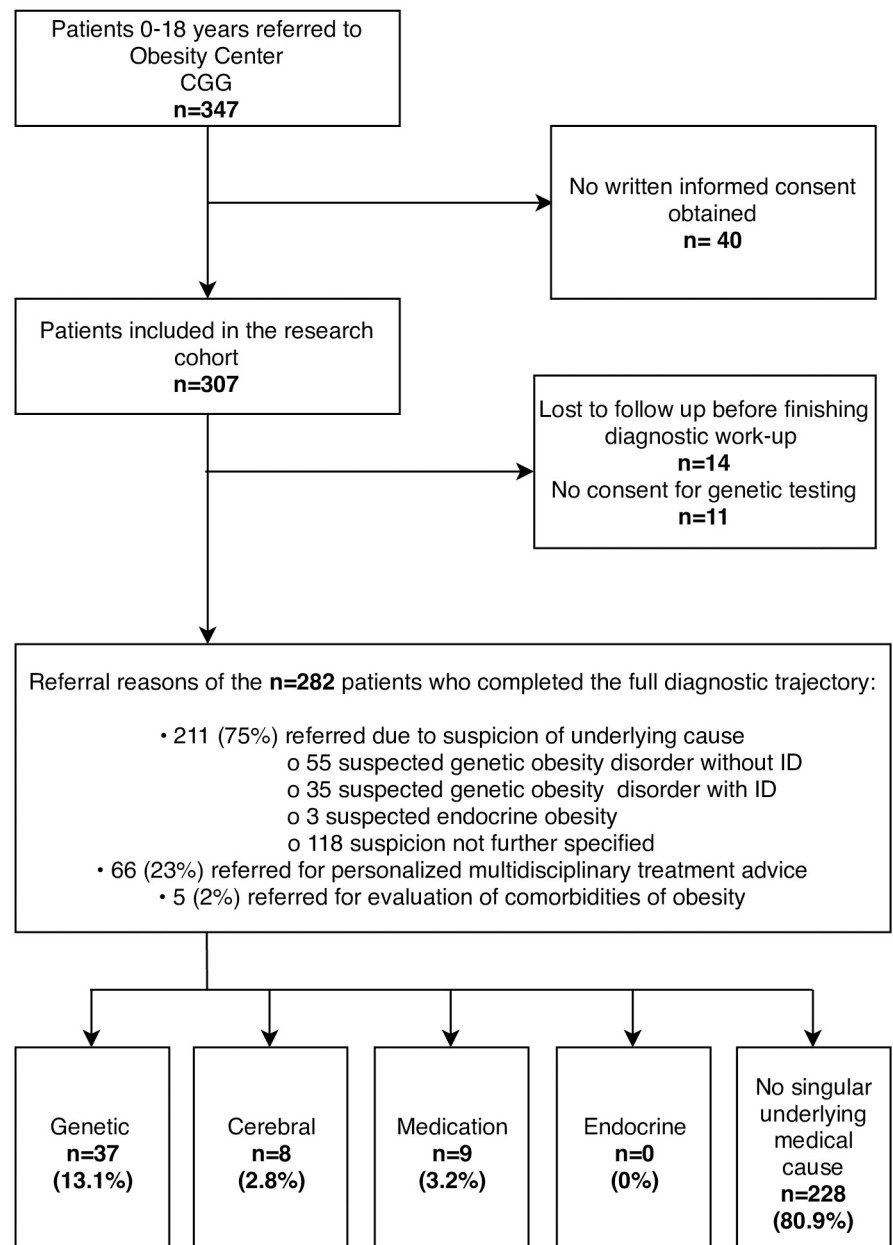

**Fig 1. Study flow chart.** Flow chart indicating the inclusion of participants and diagnoses established in our cohort. Abbreviations: CGG, Dutch: *Centrum Gezond Gewicht*; English: *Centre for Healthy Weight*; ID, intellectual disability.

by a clinical geneticist at the outpatient clinic. Patients who visited the academic center were also seen by a pediatric physiotherapist, pedagogist, and pediatric dietician. Additional diagnostics (i.e., further genetic testing, neuropsychological or radiologic assessments) were performed when clinically indicated following international clinical guidelines. After the diagnostic procedure, it was assessed for each patient whether an endocrine, genetic, cerebral or medication-induced main underlying cause of obesity could be diagnosed. Contributing factors to weight gain (e.g. sleep deprivation, screen time) were not considered as main

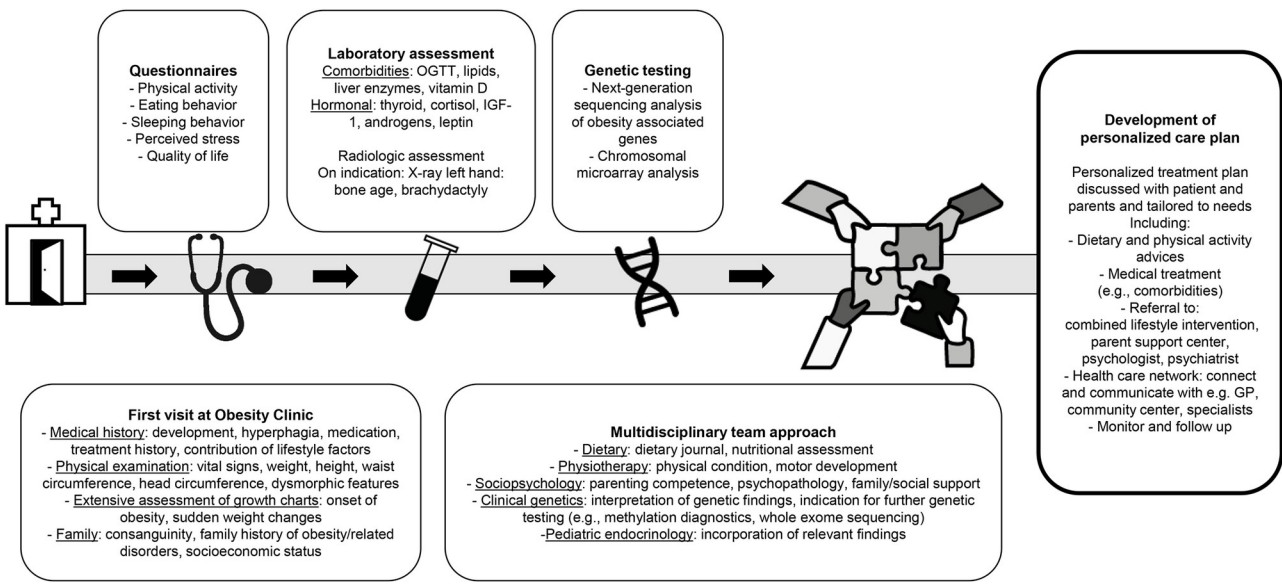

**Fig 2. Diagnostic approach.** Systematic diagnostic approach for children and adolescents with obesity and a suspicion of an underlying medical cause. Abbreviations: OGTT, oral glucose tolerance test; IGF-1, Insulin-like growth factor 1; GP, general practitioner.

underlying causes of obesity. After the diagnostic workup, a patient-tailored treatment plan was designed by the multidisciplinary team in which all relevant findings were incorporated, including advice regarding diet and physical activity, medical treatment (regarding comorbidities) or referral to combined lifestyle intervention, parent support center, psychologist, or psychiatrist. This personalized treatment plan was discussed with the patient and parents and tailored to their personal situation and needs.

## Assessments

The features that were assessed during the diagnostic workup are summarized below (details in the S1 Appendix).

**Phenotypic features.** Clinical history-taking and physical examinations were performed following the Dutch pediatric obesity guideline, including evaluation of neonatal feeding, weight-inducing medication use, development, dysmorphic features, or congenital anomalies. [17] Height, weight and head circumference were measured rounded to the nearest decimal. The Dutch national growth charts, which use the definition of pediatric obesity by Cole *et al.*, were used to calculate standard deviation scores (SDS). [3, 18] Severe obesity was defined by the IOTF definition as a BMI ≥ the age- and sex-specific IOTF BMI-values corresponding to a BMI of 35 kg/m$^2$ at age 18 years. [3] Each patient's growth charts were studied in detail to determine the age of onset of obesity and to evaluate the presence of sudden weight changes. If sudden weight changes were present, it was determined whether these changes were associated with cerebral injury (e.g., tumor in the hypothalamic region) or use of known weight-inducing medication. Short stature was defined as a height-for-age z-score <2 SDS or height-for-age <-1.6 SDS compared to target height; [19] tall stature as a height-for-age z-score >2 SDS or height-for-age >2 SDS compared to target height. [20] Intellectual disability was determined by the DSM-5 (Diagnostic and Statistical Manual of Mental Disorders 5) definition of intellectual disability or an IQ score ≤70. Family histories of bariatric surgery and extreme obesity

(BMI > 40 kg/m² for adults, or corresponding pediatric value) [3] were obtained for the past three generations. Information on consanguinity was obtained from questionnaires and additionally from the regions of homozygosity identified by SNP microarray analysis (see below). Presence of hyperphagia was determined by the physician, based on the child's or parents' answers regarding hunger, e.g., satiation and satiety, preoccupation with food, night eating, secret eating, food-seeking behavior, and the distress that accompanies the child's hunger or obsession with food. [21] Patients were considered Dutch if patient and both parents were born in The Netherlands; otherwise, patients were classified as having a migration background. [22] Presence of psychosocial/psychiatric problems was defined as the presence of an established DSM-5 diagnosis (with the exception of intellectual disability) or social problems for which official authorities were involved, such as child protective services. Additionally, Dutch neighborhood socioeconomic status z-scores were calculated. These summarize average income, education and unemployment in postal code areas to provide an estimate of the socioeconomic status of patients. [23] Finally, the contribution of lifestyle factors was assessed. As lifestyle factors play a role in every case of obesity, the multidisciplinary team determined if lifestyle factors were the most important contributor to the obesity for each patient without an underlying medical diagnosis. For example, this label determination was used for patients without an underlying medical diagnosis who reported that obesity started during the divorce of their parents and consequently never resolved. This was subsequently objectified in their growth charts.

**Laboratory assessment.** Laboratory assessment was performed for all patients. These consisted of screening for comorbidities of obesity, including standard oral glucose tolerance test, lipids, liver enzymes, vitamin D status and hormonal assessment, i.e., thyroid hormones, cortisol, insulin-like growth factor 1, androgens, and leptin. Further details are provided in the S1 Appendix.

**Genetic testing.** Obesity gene panel sequencing and single nucleotide polymorphism (SNP) microarray analysis were performed in a diagnostic setting for all patients. Three diagnostic obesity gene panel tests successively became available in The Netherlands during the time span of the study (S1 Appendix). All patients were tested at least for the most important genetic obesity disorders mentioned in the ES guideline, such as *GNAS*, *LEP*, *LEPR*, *MC4R*, *PCSK1*, *POMC*, and *SIM1*. [13] Details and complete gene lists are provided in the S1 Appendix. Obesity gene panel sequencing was performed in the ISO 15189 accredited genetic diagnostics laboratories of Amsterdam UMC and UMC Utrecht. Chromosomal microarray analysis and additional diagnostic tests were also performed at the ISO 15189 genetic diagnostics laboratories of other Dutch academic centers. Identified variants were compared with in-house and public databases to exclude common variants. Variants were classified according to the American College of Medical Genetics and Genomics (ACMG) guideline. [24] Family segregation studies were performed if necessary to clarify the pathogenicity of a variant of uncertain significance (VUS) or copy number variation (CNV). Interpretation of found variants was performed in a diagnostic setting according to the ACMG guideline. Variants of uncertain significance were not classified as genetic obesity disorder, but as a VUS/CNV that possibly explains the obesity phenotype, for which functional studies or other evidence for pathogenicity are necessary. All patients were evaluated by a clinical geneticist specialized in genetic obesity disorders to see whether further genetic testing (e.g., Prader-Willi syndrome (PWS) and Temple syndrome diagnostics, whole exome sequencing) was warranted, for example in case of unexplained intellectual disability, short stature, neonatal hypotonia, multiple congenital anomalies or other signs and symptoms of genetic obesity disorders as mentioned in the ES guideline. [13]

**Definition of underlying medical causes of obesity.** We used the following definitions of main underlying medical causes of obesity:

Genetic obesity was diagnosed when genotyping revealed known pathogenic variants in obesity-associated genes which matched the clinical phenotype. Likely pathogenic variants, as defined by the American College of Medical Genetics and Genomic (ACMG) guideline, [24] were only considered as causative if the clinical phenotype of the patient matched with the found genotype (according to the clinical features mentioned in the ES guideline) [13] and segregation analysis was indicative as well. For genetic obesity disorders not mentioned in the ES guideline, the typical phenotype was based on literature review. [25–32]

Endocrine obesity: Cushing's syndrome and clinical hypothyroidism were considered endocrine causes of obesity. Additional diagnostics for Cushing's syndrome were performed in the presence of impaired growth velocity coinciding with sudden weight gain, Cushingoid phenotype features, and abnormal laboratory results. [13, 33]

Cerebral injury was diagnosed as the cause of obesity in the presence of CNS injury affecting the hypothalamic centers for weight regulation due to craniopharyngioma surgery, meningitis or ischemic damage, coinciding with a sudden progression of obesity (seen as a clear visual slope discontinuity in the growth curve from the time of CNS injury onwards) and the absence of other plausible explanations for the sudden weight gain.

Medication-induced obesity was diagnosed in the presence of start or intensification of known weight-inducing medication (i.e., corticosteroids, anti-epileptic, anti-depressant and anti-psychotic drugs) [34–38] coinciding with a sudden progression of obesity (seen as a clear visual slope discontinuity in the growth curve) and the absence of other plausible explanations for the sudden weight gain.

## Analysis

Statistical analysis was performed using SPSS version 24.0 [IBM Corp. Armonk, NY]. Data are presented as median (interquartile range; IQR) and maximum, or mean (standard deviation; SD) and maximum, as appropriate. Differences in features between patients with genetic obesity disorders and patients without a singular underlying medical cause of obesity were analyzed using the chi-squared test, Fisher's exact test, independent sample t-test or Mann-Whitney U test, as appropriate. Two-sided p-values <0.05 were considered statistically significant, as we interpreted these comparisons as hypothesis-generating. For the same reason, we decided not to perform formal statistical testing for comparisons between other patient subgroups due to the small subgroup sizes.

## Results

### Patient characteristics

In total, 347 patients were referred to Obesity Center CGG during the time span of this study (Fig 1). Of these patients, 282 patients underwent the complete diagnostic workup and were included in these analyses. The majority of these patients presented at the academic hospital (222; 78.7%). Most patients were referred because of suspicion of an underlying cause (Fig 1). All 282 patients underwent the described gene panel analysis and chromosomal microarray analysis. After consulting with a clinical geneticist, additional genetic diagnostics were performed for 77 patients. The most important modalities were PWS diagnostics in 31 patients; whole exome sequencing in 27 patients; maternal UPD14 diagnostics in 21 patients. Median BMI for age was +3.7 SDS (IQR +3.3-+4.3), indicating severe obesity (Table 1). Most patients were Dutch (183/282, 64.9%); 99/282 (35.1%) had a migration background. In 67/282 (23.8%) of the patients intellectual disability (ID) was present.

**Table 1. Group characteristics of the study population.**

| | | All patients | Genetic obesity disorders | | | Cerebral obesity | Medication-induced obesity | No definite singular underlying medical diagnosis |
|---|---|---|---|---|---|---|---|---|
| | | Total group | Genetic obesity disorders without ID | Genetic obesity disorders with ID | Total group | Total group | Total group | Total group |
| | | n = 282 | n = 19 | n = 18 | n = 37 | n = 8 | n = 9 | n = 228 |
| *Patient characteristics* | | | | | | | | |
| Age at initial visit | Median (IQR) [max] | 10.8 (7.7–14.1) [18.0] | 10.0 (2.9–14.6) [17.7] | 11.2 (7.1–14.7) [16.3] | 10.0 (6.0–14.6) [17.7] | 11.9 (10.3–16.6) [17.5] | 12.3 (9.1–14.8) [17.3] | 10.7 (7.7–13.6) [18.0] |
| Female | n (%) | 165 (59%) | 14/19 (74%) | 12/18 (67%) | 26/37 (70%) | 5/8 (63%) | 5/9 (56%) | 129/228 (57%) |
| Early-onset <5 years | n (%) | 182 (65%) | 18/19‡ (95%) | 12/18 (67%) | 30/37‡ (81%) | 4/8 (50%) | 4/9 (44%) | 146/228 (64%) |
| Hyperphagia | n (%) | 113 (40%) | 15/19‡ (79%) | 9/18 (50%) | 24/37‡ (65%) | 2/8 (25%) | 3/9 (33%) | 84/228 (37%) |
| *Anthropometric features* | | | | | | | | |
| Height SDS | Mean (SD) [max] | +0.5 (1.3) [+4.2] | +1.1 (1.4) [+4.2] | -0.4‡ (1.3) [+1.5] | +0.3 (1.5) [+4.2] | -0.3 (0.5) [+0.3] | -0.3 (0.7) [+1.5] | +0.6 (1.3) [+3.7] |
| Weight SDS | Mean (SD) [max] | +3.7 (1.2) [+7.1] | +4.6† (1.5) [+7.0] | +2.3‡ (1.5) [+5.2] | +3.5 (1.9) [+7.0] | +3.4 (1.0) [+4.7] | +3.4 (0.5) [+4.1] | +3.8 (1.1) [+7.1] |
| BMI SDS | Median (IQR) [max] | +3.7 (+3.3 - +4.3) [+8.9] | +4.2 (+3.5 - +4.7) [+8.9] | +3.1‡ (+2.4 - +3.5) [+5.5] | +3.5 (+2.8 - +4.4) [+8.9] | +3.4 (+3.2 - +4.2) [+5.5] | +3.7 (+3.4 - +4.0) [+4.2] | +3.8 (+3.3 - +4.3) [+6.6] |
| *Other clinical features* | | | | | | | | |
| Head circumference SDS | Mean (SD) [max] | +1.4 (1.2) [+4.9] | +2.0 (1.2) [+3.9] | +0.9 (1.5) [+3.8] | +1.4 (1.5) [+3.9] | +0.8 (1.0) [+2.1] | +0.2 (1.0) [+0.8] | +1.4 (1.1) [+4.9] |
| History of neonatal feeding problems | n (%) | 17 (6%) | 0/19 | 5/18‡ (28%) | 5/37 (14%) | 1/8 (13%) | 0/9 | 11/228 (5%) |
| Autism | n (%) | 37 (13%) | 1/19 (5%) | 2/18 (11%) | 3/37 (8%) | 0/8 | 2/9 (22%) | 32/228 (14%) |
| Parents with obesity | n (%) | 190 (67% of which 68 both) | 10/19 (53%) of which 1 both | 9/18 (50%) | 19/37‡ (51%) of which 1 both | 3/8 (38%) | 7/9 (77% of which 1 both) | 161/228 (70%) of which 66 both |
| Parents with history of bariatric surgery | n (%) | 34 (12% of which 3 both) | 1/19 (5%) 1 M | 1/18 (6%) 1 M | 2/37 (5%) | 0/8 | 2/9 (22%) | 30/228 (13%) of which 3 both |
| Consanguinity | n (%) | 24 (9%) | 2/19 (11%) | 0/18 | 2/37 (5%) | 1/8 (13%) | 1/9 (11%) | 20/228 (9%) |
| Psychosocial problems | n (%) | 130 (46%) | 3/19‡ (16%) | 4/18† (22%) | 7/37‡ (19%) | 3/8 (38%) | 5/9 (56%) | 115/228 (50%) |
| Current/past use of weight-inducing medication | n (%) | 78 (28%) | 5/19 (26%) | 2/18 (11%) | 7/37 (19%) | 3/8 (38%) | 9/9 (100%) | 59/228 (26%) |
| Evidently dysmorphic appearance and/or congenital anomaly | n (%) | 49 (17%) | 1/19 (5%) | 12/18‡ (67%) | 13/37‡ (35%) | 1/8 (13%) | 3/9 (33%) | 32/228 (11%) |
| Lifestyle factors as most important contributor to obesity | n (%) | 75 (27%) | 1/19† (5%) | 0/18‡ | 1/37‡ (3%) | 0/8 | 2/9 (22%) | 72/228 (32%) |
| Socio-economic status z-score | Median (IQR) [min] | -0.1 (-1.2 - +0.5) [-4.8] | 0.0 (-1.0 - +0.5) [-2.6] | -0.3 (-1.2 - +0.3) [-1.8] | 0.0 (-1.0 - +0.4) [-2.6] | -0.2 (-1.1 - +1.1) [-3.5] | -0.4 (-1.3 - -0.4) [-3.3] | -0.1 (-1.4 - +0.5) [-4.8] |
| Short stature | n (%) | 11 (4%) | 0/19 | 4/18‡ (22%) | 4/37 (11%) | 0/8 | 0/9 | 7/228 (3%) |
| Tall stature | n (%) | 60 (21%) | 6/19 (32%) | 1/18 (6%) | 7/37 (19%) | 0/8 | 0/8 | 53/228 (22%) |

ID, intellectual disability; VUS, variant of unknown significance; CNV, copy number variation; VUS, variants of uncertain significance; IQR, interquartile range; max, maximum; SD(S), standard deviation (score); BMI, body mass index; min, minimum.

† P<0.05 versus no definite singular underlying medical diagnosis group;

‡ P<0.01 versus no definite singular underlying medical diagnosis group.

## Underlying medical causes of obesity

An underlying medical cause of obesity was identified in 54/282 (19.1%) patients in our cohort: 37 genetic obesity disorders, 9 medication-induced obesities, and 8 obesities due to cerebral injury (Table 1). None of the patients' obesity was explained by clinical hypothyroidism or Cushing's disease. In the remaining 228/282 (80.9%) patients no singular underlying medical cause of obesity could be identified. In 17 of these 228 patients a VUS/CNV [24] was identified that possibly explains the obesity phenotype, but this still requires further research, such as functional studies, and therefore falls beyond the scope of this article.

## Genetic causes

Of the 37 patients with genetic obesity, 18 patients had a genetic obesity disorder with ID, and 19 without ID. Pathogenic variants in *MC4R* were the most commonly found genetic obesity disorder in our cohort and were found in 9/37 patients, corresponding to 3.2% of the total cohort of 282 patients. The second frequently identified genetic obesity disorders were biallelic *LEPR* pathogenic variants (6/37), followed by *GNAS* pathogenic variants leading to pseudohypoparathyroidism type 1a (5/37). The specific genetic aberrations are presented in Table 2. The clinical phenotypes of all patients with genetic obesity are described in Tables 3 and 4. Although most patients with a genetic obesity disorder had a combination of clinical features typical of their genetic obesity disorder, most patients did not have the *complete* clinical phenotype as mentioned in the ES guideline (Tables 3 and Table 4). Most notably, 6 out of 18 patients who were diagnosed with a genetic obesity disorder that is typically associated with ID did not have ID or developmental delay (Table 3).

In 3/37 cases, a heterozygous mutation/CNV was identified (in 2 patients in *POMC* and in 1 patient in *PCSK1*), which constitutes important genetic risk factors for early-onset obesity as demonstrated in association studies, [27, 39] in contrast to their autosomal recessive forms which cause a more severe clinical phenotype (S1 Appendix).

## Cerebral injury as cause of obesity

We identified cerebral injury as the underlying medical cause of obesity in 8/282 (3%) patients. In five patients onset of rapid weight gain, objectified through analysis of their growth charts, coincided with intracranial surgery and/or radiotherapy (two craniopharyngiomas and three malignancies in the hypothalamic region). One patient had congenital anatomic midline defects in the hypothalamic region and clear hyperphagia and excessive weight gain from birth. In the remaining two patients onset of rapid weight gain occurred after meningitis or ischemic infarction, suggesting hypothalamic dysfunction.

## Use of known weight-inducing medication as cause of obesity

In 9/282 patients (3%) medication-induced obesity was diagnosed through the combination of extensive evaluation of their growth charts and medication history and exclusion of endocrine, genetic, or cerebral causes of obesity. Of these nine patients, six were chronic users of inhalation corticosteroids (ICS). In 5/6 patients, periods of sudden weight gain, as seen on their growth charts, coincided with intermittent use of oral corticosteroids in the absence of other plausible causes of their sudden weight gain. In the remaining patient periods of intensification of chronic ICS use coincided with sudden weight gain according to the growth chart, without other plausible explanations for the sudden weight gain. In the other three patients the start and restart of antipsychotic drugs in one, and antiepileptic drugs in two patients, coincided with sudden weight gain.

**Table 2. Overview of genetic alterations in patients diagnosed with a genetic obesity disorder.**

| Patient | Gene/CNV | Reference transcript | Genetic alteration | Inheritance |
|---|---|---|---|---|
| *Genetic obesity disorders without ID* | | | | |
| 1 | *MC4R* | NM_005912.2 | Heterozygous c.105C>A p.(Tyr35*) | M |
| 2 | *MC4R* | NM_005912.2 | Homozygous c.216C>A p.(Asn72Lys) | n.p. |
| 3 | *MC4R* | NM_005912.2 | Heterozygous c.105C>A p.(Tyr35*) | M |
| 4 | *MC4R* | NM_005912.2 | Compound heterozygous c.446_450del p.(Phe149Tyrfs*9), c.644T>G p.(Met215Arg) | P and M both heterozygous |
| 5 | *MC4R* | NM_005912.2 | Homozygous c.779C>A p.(Pro260Gln) | P and M both heterozygous |
| 6 | *MC4R* | NM_005912.2 | Heterozygous c.913C>T p.(Arg305Trp) | *de novo* |
| 7 | *MC4R* | NM_005912.2 | Heterozygous c.380C>T p.(Ser127Leu) | P |
| 8 | *MC4R* | NM_005912 | Heterozygous c.750_751del p.(Ile251Trpfs*34) | n.p. |
| 9 | *MC4R* | NM_006147.2 | Homozygous c.785del p.(Phe262Serfs*4) | n.p. |
| 10 | *LEPR* | NM_001003679.3 | Compound heterozygous c.2168c>T p.(Ser723Phe), c.1985T>C p.(Leu662Ser) | P and M both heterozygous |
| 11 | *LEPR* | NM_001003679.3 | Compound heterozygous c.2051A>C p.(His684Pro), c.2627C>A p.(Pro876Gln) | P and M both heterozygous |
| 12 | *LEPR* | NM_002303.5 | Compound heterozygous c.1753-1dup p.?, c.2168C>T p.(Ser723Phe) | P and M both heterozygous |
| 13 | *LEPR* | NM_002303.5 | Homozygous c.1604-8A>G p.? intronic pathogenic variant affecting splicing | P and M both heterozygous |
| 14 | *LEPR* | NM_002303.5 | Homozygous c.3414dup p.(Ala1139Cysfs*16) | P and M both heterozygous |
| 15 | *LEPR* | NM_002303.5 | Compound heterozygous c.1835G>A p.(Arg612His), c.2051A>C p.(His684Pro) | P and M both heterozygous |
| 16 | *PCSK1* | NM_000439.4 | Heterozygous c.541T>C p.(Tyr181His)[a] | M |
| 17 | *POMC* | NM_001035256.1 | Heterozygous c.706C>G p.(Arg236Gly)[a] | n.p. |
| 18 | *SIM1* | n/a | 6q16.3 deletion (chr6:100.879.864–102.471.598), disrupting *SIM1* | *de novo* |
| 19 | *STX16* (PHP 1b) | NM_003763.5 | Heterozygous microdeletion c.331-?_585 + ? p.? | M |
| *Genetic obesity disorders with ID* | | | | |
| 1 | *GNAS* (PHP1a) | NM_001077488 | Heterozygous c.85C>T p.(Gln29*) | M |
| 2 | *GNAS* (PHP1a) | NM_000516.4 | Heterozygous c.794G>A p.(Arg265His) | M |
| 3 | *GNAS* (PHP1a) | NM_018666.2 | Heterozygous c.665T>C p.(Met222Thr)[b] | M and PM |
| 4 | *GNAS* (PHP1a) | NM_018666.2 | Heterozygous c.665T>C p.(Met222Thr)[b] | M and PM |
| 5 | *GNAS* (PHP1a) | NM_018666.2 | Heterozygous c.665T>C p.(Met222Thr)[b] | M and PM |
| 6 | 16p11.2del | n/a | Distal 16p11.2 deletion (chr16:28,825,605–29,043,450, incl. *SH2B1*) | P and MP |
| 7 | 16p11.2del | n/a | Distal 16p11.2 deletion (chr16:28,819,029–29,043,973,incl. *SH2B1*) | *de novo* |
| 8 | 16p11.2del | n/a | Proximal 16p11.2 deletion (chr16:29,563,985–30,107,008, not incl. *SH2B1*) | *de novo* |
| 9 | mUPD14 (Temple syndrome) | n/a | Temple syndrome (caused by maternal uniparental disomy chromosome 14) | n/a |
| 10 | mUPD14 (Temple syndrome) | n/a | Temple syndrome (caused by maternal uniparental disomy chromosome 14) | n/a |
| 11 | *Epigenetic error chr14* (Temple syndrome) | n/a | Temple syndrome (caused by imprinting defect on chromosome 14) | n/a |
| 12 | *Epigenetic error chr14* (Temple syndrome) | n/a | Temple syndrome (caused by imprinting defect on chromosome 14) | n/a |
| 13 | *MKKS* (Bardet-Biedl syndrome) | NM_018848.3 | Compound heterozygous c.110A>G p.(Tyr37Cys), c.950_960del p.(Gly317Aspfs*6) | P and M both heterozygous |
| 14 | *IFT74* (Bardet-Biedl syndrome) | NM_025103.3 | Compound heterozygous c.371_372del p.(Gln124Argfs*9), c.16850-1G>T p.? | P and M both heterozygous |
| 15 | *MYT1L* | NM_015025.2 | Heterozygous c.808del p.(Gln270Lysfs*11) | *de novo* |
| 16 | *POMC* | n/a | 2p deletion (chr2:22.791.486–27.942.764), containing *POMC* | *de novo* |

(*Continued*)

**Table 2.** (Continued)

| Patient | Gene/CNV | Reference transcript | Genetic alteration | Inheritance |
|---|---|---|---|---|
| **17** | *SPG11* (Spastic paraplegia 11) | NM_025137.3 | Compound heterozygous c.4534dup p.(Asp1512Glyfs*7), c.5867-?_6477+? del p.? (deletion of exons 31–34 | P and M both heterozygous |
| **18** | *VPS13B* (Cohen syndrome) | NM_017890.4 | Compound heterozygous c.2911C>T p.(Arg971*), c.8697-2A>G p.? | P and M both heterozygous |

CNV, copy number variation; SDS, standard deviation score; BMI, body mass index in kg/m2; ID, intellectual disability; M, mother; P, father; n.p., segregation analysis not performed; PHP 1b, pseudohypoparathyroidism type 1b; PHP 1a, pseudohypoparathyroidism type 1a; PM, father of mother; MP, mother of father; n/a, not applicable.

[a]important genetic risk factor contributing to severe early-onset obesity;

[b]siblings.

## Comparison of phenotype in patients with genetic obesity disorders and patients without a singular underlying medical cause of obesity

Patients with genetic obesity disorders more often had an extreme early-onset of obesity <5 years (p = 0.04) and hyperphagia (p = 0.001) when compared to patients without a singular underlying medical cause of obesity (Table 1, detailed p-values in S1 Table). Furthermore, the presence of obesity in parents (p = 0.02) and psychosocial problems (determined by the involvement of official authorities or DSM-V diagnosis; p = = 0.001) were less often present in the genetic obesity group. No significant differences were found with respect to BMI SDS, sex, socio-economic status z-score and family history of consanguinity or bariatric surgery (all p>0.05; detailed p-values in S1 Table). When zooming in on patients with genetic obesity with ID, they more often had short stature (p = 0.005), a history of neonatal feeding problems (p = 0.003), a dysmorphic appearance and/or congenital anomalies (p<0.001), and less severe obesity (lower BMI SDS; p<0.001) than patients without a singular underlying medical cause of obesity. Extreme early-onset obesity <5 years and hyperphagia were not present more often in the patients with genetic obesity disorders with ID (Table 1). With regard to height SDS, patients with genetic obesity without ID had a higher height SDS than patients without a singular underlying medical cause of obesity, although this difference was not statistically significant (p = 0.19). In contrast, patients with genetic obesity with ID had a significantly lower height SDS (p = 0.004).

## Comparison of patients with cerebral or medication-induced obesities with other subgroups of patients

No assessed phenotype features were specifically present or absent in patients with cerebral or medication-induced obesities (Table 1). However, on a group level, these patients had lower height SDS than patients with genetic obesity disorders without ID or patients without underlying medical causes of the obesity.

## Discussion

In this study, an extensive systematic diagnostic approach in a specialized obesity center established an underlying medical cause of obesity in 19% of pediatric patients. These included genetic obesity disorders (13%), medication-induced obesities (3%) and obesities due to cerebral injury (3%). To the best of our knowledge, this is the first study which reports the yield of a broad diagnostic workup in a tertiary pediatric obesity cohort, focusing not only on genetic obesity disorders but also on endocrine, medication-induced, and cerebral causes of obesity.

**Table 3. Clinical characteristics of patients diagnosed with a genetic obesity disorder with ID.**

| Gene/CNV | GNAS (PHP1a) | 16p11.2 deletion syndrome | Temple syndrome | MKKS (Bardet-Biedl syndrome) | IFT74 (Bardet-Biedl syndrome) | MYT1L | 2p-deletion syndrome | SPG11 (Spastic paraplegia 11) | VPS13B (Cohen syndrome) |
|---|---|---|---|---|---|---|---|---|---|
| Genetic cause* | Heterozygous disease-associated variant | 16p11.2 deletion | Maternal uniparental disomy or imprinting defect of chromosome 14 | Compound heterozygous disease-associated variants | Compound heterozygous disease-associated variants | Heterozygous disease-associated variant | 2p-deletion syndrome, incl. POMC | Compound heterozygous disease-associated variants | Compound heterozygous disease-associated variants |
| Number of patients | 5 | 3 | 4 | 1 | 1 | 1 | 1 | 1 | 1 |
| Age at diagnosis in years, median (range) | 11.6 (3.7–14.8) | 6.6 (4.2–15.3) | 9.8 (5.0–15.1) | 1.7 | 11.2 | 3.3 | 12.8 | 14.0 | 4.4 |
| *Clinical features at initial visit* | | | | | | | | | |
| Age in years, range | 3.7–14.8 | 4.2–15.8 | 8.1–15.1 | 4.6 | 8.9 | 5.5 | 14.6 | 11.2 | 8.5 |
| Height SDS, median (range) | -1.0 (-2.2 – -0.5) | +0.9 (-2.4 – +1.5) | -1.0 (-2.1 – +1.1) | +0.7 | +1.5 | -0.6 | -1.2 | +1.4 | -0.7 |
| Δ Height SDS vs target height SDS, median (range) | -0.6 (-2.1 – +0.8) | +0.9 (-0.7 – +1.6) | -1.1 (-2.2 – +1.6) | +0.3 | +0.9 | 0.0 | 0.0 | +2.3 | -0.7 |
| BMI, median (max) | 20.9 (27.1) | 29.4 (30.1) | 31.2 (33.4) | 25.2 | 24.6 | 19.6 | 32.5 | 27.7 | 20.6 |
| BMI SDS, median (max) | +1.8 (+3.6) | +2.8 (+5.3) | +3.3 (+3.5) | +5.5 | +3.0 | +2.5 | +3.3 | +3.4 | +2.6 |
| Early-onset <5 years | 5/5 | 1/3 | 2/4 | Yes | No | Yes | Yes | Yes | No |
| Hyperphagia | 1/5 | 2/3 | 3/4 | No | No | Yes | Yes | Yes | No |
| ID | 5/5 | 1/3 | 1/4 | Too young for formal testing; not suspected | No | Yes | Yes | Yes | Yes |
| History of abnormal neonatal feeding behavior | No | No | Hypotonia/feeding problems 4/4 | Reduced satiety 1/1 | Reduced satiety 1/1, resolved after infancy | No | No | No | Hypotonia/feeding problems 1/1 |
| Clinical features characteristic of the genetic obesity disorder as mentioned in the Endocrine Society Guideline | Short stature in some but not all patients 1/5 | Hyperphagia 2/3 | *Genetic obesity syndrome not mentioned in guideline* | Developmental delay 1/1 | Developmental delay 0/1 | *Genetic obesity syndrome not mentioned in guideline* | *Genetic obesity syndrome not mentioned in guideline* | *Genetic obesity syndrome not mentioned in guideline* | *Genetic obesity syndrome not mentioned in guideline* |
| | Skeletal defects[b] 4/5 | Disproportionate hyperinsulinemia 0/3 | | Dysmorphic extremities[c] 1/1 | Dysmorphic extremities[c] 1/1 | | | | |
| | Impaired olfaction 0/5 | Early speech and language delay 2/3 that often resolves 0/3 | | Retinal dystrophy or pigmentary retinopathy 1/1 | Retinal dystrophy or pigmentary retinopathy 1/1 | | | | |
| | | | | Hypogonadism n/a (due to young age) | Hypogonadism - | | | | |
| | Hormone resistance (e.g., parathyroid hormone) 5/5 | Behavioral problems including aggression 0/3 | | Renal abnormalities/impairment 1/1 | Renal abnormalities/impairment 0/1 | | | | |

*(Continued)*

**Table 3.** (Continued)

| Gene/CNV | GNAS (PHP1a) | 16p11.2 deletion syndrome | Temple syndrome | MKKS (Bardet-Biedl syndrome) | IFT74 (Bardet-Biedl syndrome) | MYT1L | 2p-deletion syndrome | SPG11 (Spastic paraplegia 11) | VPS13B (Cohen syndrome) |
|---|---|---|---|---|---|---|---|---|---|
| **Additional clinical features characteristic of the genetic obesity syndrome** | Subcutaneous calcifications 1/5; Round facies 3/5 | N/A | Neonatal hypotonia 4/4; Neonatal feeding difficulties 4/4; Short stature 2/4; Precocious puberty 4/4; Mild intellectual disability 2/4 | N/A | N/A | ID 1/1; Autism 0/1; Behavioral problems 0/1 | Clinical features depending on size and location of deletion including hyperphagia (1/1). Generally no proopiomelanocortin deficiency (0/1). Additionally in our patient: ID, coarse facies with large front teeth | Progressive spastic paraplegia 1/1; ID 1/1; Peripheral neuropathy 0/1 | Failure to thrive in childhood 1/1; Hypotonia 1/1; Microcephaly 1/1; Visual impairment 1/1; Neutropenia 1/1; Prominent central incisors/uplifted upper lip 1/1 |
| **Presence of genetic alteration in parents** | All inherited from mother | 1 inherited from father, *2 de novo* | N/A | Both parents heterozygous | Both parents heterozygous | *De novo* | *De novo* | Both parents heterozygous | Both parents heterozygous |
| **Presence of obesity in parents who carry the genetic alteration** | Obesity *not* present | Obesity *not* present | N/A | Obesity *not* present (not associated with heterozygosity) | Obesity present in father (not associated with heterozygosity) | N/A | N/A | Obesity present in mother (not associated with heterozygosity) | Obesity present in father (not associated with heterozygosity) |

CNV, copy number variation; SDS, standard deviation score; BMI, body mass index; ID, intellectual disability; N/A, not applicable; PTH, parathyroid hormone; TSH, thyroid-stimulating hormone.

*exact genetic alterations are listed in Table 2.

[a]history of abnormal neonatal feeding behavior, i.e. reduced satiety and/or hypotonia/feeding problems;

[b]skeletal defects, i.e. short metacarpalia dig IV and V (hands and/or feet);

[c]dysmorphic extremities, e.g. syndactyly/brachydactyly/polydactyly, in our patients polydactyly.

**Table 4. Clinical characteristics of patients diagnosed with a genetic obesity disorder without ID.**

| Gene/CNV | *MC4R* | | *LEPR* | *POMC* | 6q16.3 deletion | *PCSK1* | *STX16* (PHP1b) |
|---|---|---|---|---|---|---|---|
| **Genetic cause[*]** | Homoyzygous/ compound heterozygous disease-associated variants | Heterozygous disease-associated variant | Homoyzygous/ compound heterozygous disease-associated variants | Heterozygous disease-associated variant | 6q16.3 deletion incl. part of *SIM1* | Heterozygous disease-associated variant | Heterozygous disease-associated variant |
| **Number of patients** | 4 | 5 | 6 | 1 | 1 | 1 | 1 |
| **Age at diagnosis in years, median (range)** | 9.2 (1.6–15.4) | 7.1 (2.2–15.3) | 3.9 (0.7–14.8) | 10.0 | 9.1 | 11.8 | 14.8 |
| *Clinical features at initial visit* | | | | | | | |
| **Age in years, range** | 6.5–15.4 | 2.5–15.3 | 0.7–17.7 | 10.0 | 9.1 | 12.2 | 17.2 |
| **Height SDS, median (range)** | +0.8 (+0.7 –+2.2) | +2.1 (0.0 –+4.2) | +1.0 (-1.2 –+2.2) | -0.2 | +3.0 | -0.2 | -0.1 |
| **Δ Height SDS vs target height SDS, median (range)** | +1.4 (+0.7 –+3.2) | +0.7 (-0.1 –+4.1) | +1.2 (-1.3 –+1.5) | -0.5 | +2.4 | +1.0 | -0.6 |
| **BMI, median (max)** | 34.0 (41.5) | 27.9 (38.6) | 35.3 (47.5) | 28.2 | 36.8 | 32.9 | 31.4 |
| **BMI SDS, median (max)** | +4.3 (+5.2) | +4.2 (+5.4) | +4.6 (+8.9) | +3.9 | +4.4 | +3.5 | +2.9 |
| **Early-onset <5 years** | 3/4 | 5/5 | 6/6 | Yes | Yes | Yes | Yes |
| **Hyperphagia** | 3/4 | 5/5 | 5/6 | No | Yes | No | Yes |
| **ID** | 0/4 | 0/4 | 0/6 | No | No | No | No |
| **History of abnormal neonatal feeding behavior** | Reduced satiety 3/4 | No | Reduced satiety 4/6 | No | No | No | Reduced satiety 1/1 |
| **Clinical features characteristic of the genetic obesity disorder as mentioned in the Endocrine Society Guideline** | Hyperphagia 4/4<br><br>Accelerated linear growth 3/4<br><br>Disproportionate hyperinsulinemia 4/4<br><br>Low/normal blood pressure 1/4 | Hyperphagia 4/5<br><br>Accelerated linear growth 3/5<br><br>Disproportionate hyperinsulinemia 1/5<br><br>Low/normal blood pressure 4/4[b] | Extreme hyperphagia 5/6<br><br>Frequent infections 0/6<br><br>Hypogonadotropic hypogonadism 3/4[c]<br><br>Mild hypothyroidism 2/6 | *Genetic obesity syndrome not mentioned in guideline* | *Genetic obesity syndrome not mentioned in guideline* | *Genetic obesity syndrome not mentioned in guideline* | *Genetic obesity syndrome not mentioned in guideline* |
| **Additional clinical features characteristic of the genetic obesity disorder** | More severe phenotype than autosomal dominant | N/A | Growth hormone deficiency 1/6 | Hyperphagia (less severe than autosomal recessive *POMC* deficiency) 0/1 | Characteristics depending on size of deletion: Intellectual disability 0/1<br><br>Autism 0/1<br><br>Behavioral problems 0/1 | Hyperphagia (less severe than autosomal recessive *PCSK1* deficiency) 0/1 | PTH resistance 1/1;<br><br>Occasionally partial TSH resistance 1/1<br><br>Enhanced intrauterine growth 1/1<br><br>Occasionally mild brachydactyly 1/1<br><br>Round facies 1/1 |

*(Continued)*

**Table 4.** (Continued)

| Gene/CNV | MC4R | | LEPR | POMC | 6q16.3 deletion | PCSK1 | STX16 (PHP1b) |
|---|---|---|---|---|---|---|---|
| **Presence of genetic alteration in parents** | 2/4 both parents heterozygous 2/4 n.p. | 3/5 inherited from parent, 1/5 *de novo*, 1/5 n.p. | All parents heterozygous | n.p. | *De novo* | Inherited from mother | Inherited from mother |
| **Presence of obesity in parents who carry the genetic alteration** | Obesity present in 1/4 heterozygous parents (known reduced penetrance) | Obesity present in 1/3 heterozygous parents (known reduced penetrance) | Obesity present in 3/12 heterozygous parents (unclear association with heterozygosity) | N/A | N/A | Obesity present in heterozygous mother | Obesity *not* present in heterozygous mother |

PHP1b, pseudohypoparathyroidism type 1b; SDS, standard deviation score; BMI, body mass index; ID, intellectual disablility; BP, blood pressure; N/A, not applicable; PTH, parathyroid hormone; TSH, thyroid-stimulating hormone; n.p., not performed.

*exact genetic alterations are listed in Table 2.

a history of abnormal neonatal feeding behavior, i.e. reduced satiety and/or hypotonia/feeding problems;

b In 1 patient, BP could not be measured due to unrest.

c In 2 prepubertal patients not (yet) detectable.

Previously, Reinehr *et al.* assessed the prevalence of endocrine causes and of specific genetic causes, namely clinically identifiable syndromal causes and *MC4R* pathogenic variants in a subgroup of their cohort. [7] Their study, performed in 1405 children and adolescents visiting a specialized clinic for endocrinology and obesity, demonstrated an underlying disorder in 13 (1.7%) patients.

There are some explanations for our high diagnostic yield. First, our patients constitute a tertiary pediatric obesity population with severe obesity who were referred because of a suspicion of an underlying medical cause, or resistance to lifestyle interventions. Thus, we had a higher *a priori* probability of finding underlying medical causes than in an unselected pediatric obesity population. Nevertheless, we show that a broad systematic diagnostic workup is needed to identify these diverse underlying causes of obesity. Secondly, medication use and cerebral/hypothalamic injury were not mentioned in the evaluation of other cohorts, although they are part of the recommended diagnostic workup of the ES guideline for pediatric obesity. [13] Furthermore, the guideline mentions only antipsychotics as weight-inducing medication, but we also considered specific antipsychotic or anti-epileptic drugs [37, 38] and prolonged use of corticosteroids [35, 36] as potential cause of obesity in individual patients, but only in the presence of a temporal relationship with onset of obesity, objectified through comprehensive growth chart analysis, and in the absence of other underlying medical causes of obesity or other plausible explanations for the sudden weight gain. Comprehensive growth chart analysis was also supportive in the identification of patients with cerebral/hypothalamic injury as the cause of their obesity in our cohort. Thus, future guidelines might benefit from adding growth chart analysis as part of the diagnostic workup of pediatric obesity. Thirdly, intellectual disability was present in 24% of patients, which increased the *a priori* probability of genetic obesity disorders with ID. The last explanation for our high yield is the extensive genetic testing we performed. Pathogenic variants in *MC4R* were the most frequently identified genetic cause of obesity in our cohort (9/282 patients, 3.2%). This number is comparable to previous findings in another Dutch tertiary pediatric cohort (2.1%) [40] and 1.6–2.6% in other non-consanguineous pediatric cohorts screening for genetic obesity. [41, 42] However, in many studies, only *MC4R* mutations or a small number of obesity-associated genes are tested. [7, 27, 40–43] In our cohort, 13 genetic obesity disorders other than *MC4R* were present. Thus, this study shows

that extensive genotyping can highly augment the diagnostic yield when performed in similar pediatric obesity cohorts. The extent to which heterozygous mutations/CNV in *PCSK1* and *POMC* are involved in monogenic obesity remains a point of discussion. Association studies clearly demonstrate that these rare variants contribute to a highly increased risk for obesity. [27, 39] Moreover, identifying these patients is of clinical importance for patient-tailored treatment as clinical trials with MC4R-agonist setmelanotide will be conducted, as it is hypothesized that these patients will have reduced MC4R functioning. [44]

We did not identify patients with an endocrine disorder as the cause of obesity. None of the patients were diagnosed with Cushing's syndrome. Pediatric Cushing's syndrome is extremely rare, [45] and patients are often referred due to impaired growth velocity and abnormal laboratory results. [13] Therefore, in contrast to adults, these patients are not primarily referred to obesity clinics. Retrospective analysis of ICD-10 codes for Cushing's syndrome in the central hospital registries at both participating centers during the entire study period (2015–2018) showed four diagnoses of pediatric Cushing's syndrome in these years; none of these four patients developed severe obesity. Importantly, PWS, the most common genetic obesity disorder with ID, was not identified in our cohort. This can be explained by the fact that in Dutch pediatric practice, PWS is often diagnosed during the neonatal period due to the typical hypotonia and feeding problems and after diagnosis, clinical care is transferred to specialized PWS expertise centers.

The second aim of our study was to present the phenotype of patients with underlying medical causes and investigate whether they can be distinguished from patients without underlying medical causes. We therefore performed the comprehensive diagnostic workup in all patients. In daily clinical practice with lower *a priori* probability of underlying medical causes, it is complex to determine for whom these diagnostics should be performed. According to literature, one of the most important features to help distinguish these patients is their stature. Reinehr *et al.* reported that short stature had a high sensitivity for underlying causes of obesity in their cohort. [7] In our study, patients with genetic obesity disorders associated with ID, and patients with cerebral and medication-induced obesities in our cohort indeed had lower height SDS than expected based on the fact that obesity is associated with taller stature. [46] However, most of these patients did not fulfill the definition for short stature. [19] Unsurprisingly, cardinal features of genetic obesity disorders, namely early onset of obesity (<5 years) and hyperphagia, were more often present in patients with genetic obesity, but only when ID was not present. On the other hand, patients with genetic obesity disorders with ID more often had a history of neonatal feeding problems and congenital anomalies or dysmorphic features. Thus, presence of these features should lead to consideration to perform additional diagnostics. Contrary to expectations BMI SDS was not significantly higher in patients with genetic obesity compared to patients without underlying medical causes. A possible explanation is that severity of obesity increases the probability of being referred to a pediatric obesity center regardless of whether genetic obesity is diagnosed. Important factors that were more frequently present in the patients without underlying medical causes were psychosocial problems (DSM-5 diagnosis or involvement of authorities such as child protective services). These psychosocial problems might contribute to developing a higher BMI SDS. [47] On group level, we did not find evidence for significant differences in socio-economic status scores between patients with genetic obesity and patients without underlying medical causes, but individual differences in socio-economic factors and obesogenic environments might also play a role. Interestingly, parents of children with a genetic obesity disorder more often had no obesity than parents of children without an underlying cause. This sounds counterintuitive for hereditary obesity disorders, but can be explained by the fact that most of the genetic aberrations in our cohort had occurred *de novo* or had an autosomal recessive inheritance pattern. Thus, negative family

history of obesity could therefore suggest a genetic obesity disorder. In conclusion, we show that several phenotypic features differed significantly between patients with and without underlying medical causes of obesity, but no feature was specific. Thus, a broad diagnostic workup is warranted in patients with a high suspicion of an underlying medical cause of obesity, e.g., in cases with early-onset obesity, hyperphagia, relatively low height SDS (especially in the presence of ID) and presence of sudden weight changes objectified through comprehensive growth chart analysis.

Treatment of multifactorial disorders such as obesity is complex. In our approach, all patients received a multidisciplinary treatment advice tailored to their personal needs, including personalized dietary and physical activity advice (Fig 2). Furthermore, a monitoring and follow-up plan was developed for every patient. Local health care providers, including child health clinic physicians, general practitioners, general pediatricians, and psychologists, were contacted for local implementation of the care plan. In cases with severe hyperphagia, parental support by an educational therapist was offered to cope with the child's behavior. Rehabilitation physicians were consulted when obesity interfered with performance of daily activities such as walking. [10]

Establishing a main underlying cause of obesity can improve personalized treatment. [34] In all our 54 patients with an underlying medical cause, counseling about the diagnosis was given. This included advice pertaining to bariatric surgery, which has unclear long-term success rates for patients with underlying medical causes. [43, 48] Patients with genetic obesity were counseled by a clinical geneticist regarding inheritance, associated medical problems and reproductive decisions. Hormonal supplementation was started in case of hormonal deficiencies associated with specific genetic obesity disorders (such as growth hormone treatment in cases with leptin receptor deficiency). [49] In cases of syndromic obesity, the patients were evaluated for associated organ abnormalities or referred for disease-specific surveillance. [13, 25–32] In patients with cerebral/hypothalamic injury as cause of obesity and hyperphagia, dexamphetamine treatment was considered. [50] In patients with medication-induced obesity, evaluation of necessity and alternatives for the weight-inducing medication took place in collaboration with the prescribing physician. Follow-up studies are necessary to evaluate the different individual responses to these treatment options. Interesting novel developments are clinical trials with MC4R-agonists in patients with leptin-melanocortin pathway deficiencies, e.g. *POMC* and *LEPR* deficiency, [44] and glucagon-like peptide 1 (GLP-1) agonists for adolescents with obesity. [51] These GLP-1 agonists might also be a future treatment option for patients with genetic obesity disorders, as they have been shown to be equally as effective in adults with heterozygous *MC4R* mutations compared to adults without. [52] Recently, it was suggested that a subgroup of patients with severe early-onset obesity might have relative leptin deficiency and therefore might benefit from recombinant leptin administration. [53] However, the (long-term) effects of these new potential treatment options remain to be investigated.

## Strengths and limitations

A major strength of our study is the use of a systematic diagnostic strategy in all patients investigating all medical causes of obesity mentioned in the current international guideline. [13] Moreover, we performed genetic diagnostics in all patients, and further genetic tests when clinically indicated. Furthermore, our relatively high diagnostic yield enabled us to describe the clinical phenotypes of a large number (n = 54) of patients with underlying causes of obesity from a relatively small patient cohort of 282 patients. When performing research in a diagnostic setting, one faces logistical limitations. During our study, three different versions of the diagnostic obesity-associated gene panel test were successively available for clinical use in The

Netherlands. Importantly, in all used gene panels at least the most important and well-known obesity-associated genes were tested, including among others *LEP*, *LEPR*, *MC4R*, *POMC*, *PCSK1*, *ALMS1*, *GNAS*, *SH2B1*, and *SIM1*. A strength of our diagnostic setting is that we followed the current ACMG guidelines for variant calling, leading to stringent selection of only pathogenic and likely pathogenic variants for which evidence from validated functional studies and from control populations has already been incorporated. [24] Children and adolescents with a high suspicion of a genetic cause with negative genetic testing results should be viewed as 'unsolved cases', for which current genetic tests are not yet able to pinpoint a diagnosis. As the field of obesity genetics is progressing rapidly, very recently discovered obesity genes were not present in the used diagnostic gene panels. [54] Incorporating these obesity genes might have resulted in an even higher diagnostic yield. Moreover, newer techniques such as whole-genome sequencing will become more easily accessible and affordable in clinical practice and will likely lead to more genetic obesity diagnoses.

We understand that our comprehensive approach is not feasible in every clinical setting, but our data suggest that it has added value for selected patient groups. Prospective studies looking at predictors for underlying medical causes of obesity are necessary but are difficult to establish because of the rarity of these disorders and overlap with common obesity. International collaboration in large multicenter studies using a similar standardized comprehensive approach are required.

## Conclusion

In conclusion, we show that a large variety of underlying medical obesity diagnoses can be established in pediatric patients with obesity in tertiary care setting when using a comprehensive diagnostic workup. Investigating endocrine, genetic, cerebral and medication-induced causes of obesity is needed for these patients to facilitate disease-specific and patient-tailored treatment. Further studies on predictors of underlying medical causes of obesity are needed to improve identification of these patients.

## Supporting information

**S1 Appendix. S1 Appendix to 'Identifying underlying medical causes of pediatric obesity: Results of a systematic diagnostic approach in a pediatric obesity center'.**
(DOCX)

**S1 Table. Dataset for 'Identifying underlying medical causes of pediatric obesity: Results of a systematic diagnostic approach in a pediatric obesity center'.**
(XLSX)

## Acknowledgments

We thank E. Hofland, A.G. van der Zwaan—Meijer, C.J.A. Jansen—van Wijngaarden, E. Koster, L. Bik, F. Jacobowitz and all participating patients and caregivers.

Author contributions: Literature search was performed by LK, OA, BvdV, BvdZ, MA, EMJB, MMvH, ELTvdA; study design by all authors except MA; data collection by LK, OA, HTMJ, AEB, BvdZ, EMJB, MMvH, ELTvdA; data analysis by LK, OA, BvdZ, MA, EMJB; data interpretation by all authors except HTMJ; generation of figures by LK, OA; writing by LK, OA, MMvH, ELTvdA; critical revision for important intellectual content by all authors.

## Author Contributions

**Conceptualization:** Lotte Kleinendorst, Ozair Abawi, Bibian van der Voorn, Mieke H. T. M. Jongejan, Jenny A. Visser, Elisabeth F. C. van Rossum, Bert van der Zwaag, Mariëlle Alders, Mieke M. van Haelst, Erica L. T. van den Akker.

**Data curation:** Lotte Kleinendorst, Ozair Abawi, Mieke H. T. M. Jongejan, Annelies E. Brandsma, Bert van der Zwaag, Mariëlle Alders, Elles M. J. Boon, Mieke M. van Haelst, Erica L. T. van den Akker.

**Formal analysis:** Lotte Kleinendorst, Ozair Abawi, Bert van der Zwaag, Mariëlle Alders, Elles M. J. Boon, Erica L. T. van den Akker.

**Funding acquisition:** Elisabeth F. C. van Rossum, Mieke M. van Haelst, Erica L. T. van den Akker.

**Investigation:** Lotte Kleinendorst, Ozair Abawi, Bert van der Zwaag, Mariëlle Alders, Elles M. J. Boon, Mieke M. van Haelst, Erica L. T. van den Akker.

**Methodology:** Lotte Kleinendorst, Ozair Abawi, Bibian van der Voorn, Mieke H. T. M. Jongejan, Annelies E. Brandsma, Jenny A. Visser, Elisabeth F. C. van Rossum, Bert van der Zwaag, Mariëlle Alders, Elles M. J. Boon, Mieke M. van Haelst, Erica L. T. van den Akker.

**Project administration:** Ozair Abawi, Mieke H. T. M. Jongejan, Annelies E. Brandsma, Elisabeth F. C. van Rossum, Mieke M. van Haelst.

**Resources:** Lotte Kleinendorst, Bibian van der Voorn, Mieke H. T. M. Jongejan, Annelies E. Brandsma, Jenny A. Visser, Elisabeth F. C. van Rossum, Bert van der Zwaag, Mariëlle Alders, Elles M. J. Boon, Mieke M. van Haelst, Erica L. T. van den Akker.

**Software:** Bert van der Zwaag, Mariëlle Alders, Elles M. J. Boon.

**Supervision:** Elisabeth F. C. van Rossum, Mieke M. van Haelst, Erica L. T. van den Akker.

**Validation:** Lotte Kleinendorst, Ozair Abawi, Bibian van der Voorn, Mieke H. T. M. Jongejan, Annelies E. Brandsma, Jenny A. Visser, Elisabeth F. C. van Rossum, Bert van der Zwaag, Mariëlle Alders, Elles M. J. Boon, Mieke M. van Haelst.

**Visualization:** Lotte Kleinendorst, Ozair Abawi, Erica L. T. van den Akker.

**Writing – original draft:** Lotte Kleinendorst, Ozair Abawi.

**Writing – review & editing:** Bibian van der Voorn, Mieke H. T. M. Jongejan, Annelies E. Brandsma, Jenny A. Visser, Elisabeth F. C. van Rossum, Bert van der Zwaag, Mariëlle Alders, Elles M. J. Boon, Mieke M. van Haelst, Erica L. T. van den Akker.

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
