## [Decision Letter · Decision Letter 0]

19 Mar 2020

PONE-D-20-05457

Identifying underlying medical causes of pediatric obesity: results of a systematic diagnostic approach in a pediatric obesity center

PLOS ONE

Dear Dr. van den Akker,

Thank you for submitting your manuscript to PLOS ONE. After careful consideration, we feel that it has merit but does not fully meet PLOS ONE’s publication criteria as it currently stands. Therefore, we invite you to submit a revised version of the manuscript that addresses the points raised during the review process.

Your manuscript has now been reviewed by two independent reviewers. As you can see from the positive reviews, both thought that this was a strong manuscript that would be of great interest to the readers of PLOS ONE. Both reviewers did have minor suggestions for the manuscript which we hope you will take into consideration, including adding additional discussion of how you treat and manage your patients. We look forward to receiving a revised manuscript with these minor changes for publication in PLOS ONE. 

We would appreciate receiving your revised manuscript by May 03 2020 11:59PM. To enhance the reproducibility of your results, we recommend that if applicable you deposit your laboratory protocols in protocols.io, where a protocol can be assigned its own identifier (DOI) such that it can be cited independently in the future. For instructions see: http://journals.plos.org/plosone/s/submission-guidelines#loc-laboratory-protocols

We look forward to receiving your revised manuscript.

Kind regards,

David A. Buchner, PhD

Academic Editor

PLOS ONE

Journal Requirements:

Reviewers' comments:

Reviewer's Responses to Questions

**Comments to the Author**

1. Is the manuscript technically sound, and do the data support the conclusions?

Reviewer #1: Yes

Reviewer #2: Yes

2. Has the statistical analysis been performed appropriately and rigorously? 

Reviewer #1: Yes

Reviewer #2: I Don't Know

3. Have the authors made all data underlying the findings in their manuscript fully available?

Reviewer #1: No

Reviewer #2: No

4. Is the manuscript presented in an intelligible fashion and written in standard English?

Reviewer #1: Yes

Reviewer #2: Yes

5. Review Comments to the Author

Reviewer #1: a nicely written straightforward audit of clinical practice of a clinical problem that will continue to require specialist input. this wil be of interest to clinical groups doing similar work and will be of great benefit to those thinking of setting up such a service .

the finding look to commensurate with what one might expect from what we know to date about causes of and give it a helpful real world feel which is to it credit

being as this group look to be doing such good work,i wonder if they might like to share a little more on how they treat and manage their patients with a know diagnosis-they mention it briefly around line 415 but their reflections on say leptin replacemenr or glp1 based therapies etc would be helpful to the target audience

Reviewer #2: This is a very important study that utilized an extensive systematic diagnostic approach in a specialized obesity center to identify underlying causes of obesity among pediatric patients. This adds significantly to our knowledge base of the etiology of pediatric obesity and demonstrates that patients being referred to specialty clinics may have a higher prevalence of genetic obesity, medication induced obesity, and obesity due to cerebral injury.

Major comments:

More exploration as to why the investigators believe that BMI SDA was not significantly higher in patients with genetic obesity disorder when compared to patients without underlying medical causes. How does the obesogenic environment and socioeconomic factors in which we live play a role? This needs more emphasis in the discussion.

More information regarding how to specifically treat the different types of obesity would be beneficial in the discussion (Though I recognize it must be brief given word constraints).

Minor comments:

There are a couple of times throughout the manuscript that the authors write, “significantly more often”. This is awkward to read and should be revised for clarity.

The manuscript would benefit from editing by a native English speaker.

6. PLOS authors have the option to publish the peer review history of their article (what does this mean?). If published, this will include your full peer review and any attached files.

Reviewer #1: No

Reviewer #2: No

---

## [Author Response · Author response to Decision Letter 0]

7 Apr 2020

Review Comments to the Author

Reviewer #1: a nicely written straightforward audit of clinical practice of a clinical problem that will continue to require specialist input. this will be of interest to clinical groups doing similar work and will be of great benefit to those thinking of setting up such a service .

the finding look to commensurate with what one might expect from what we know to date about causes of and give it a helpful real world feel which is to it credit

being as this group look to be doing such good work, i wonder if they might like to share a little more on how they treat and manage their patients with a known diagnosis-they mention it briefly around line 415 but their reflections on say leptin replacement or glp1 based therapies etc would be helpful to the target audience

Authors’ response: We would like to thank the reviewer for their thoughtful comments. Based on his/her comment, as well as the comment from reviewer #2, we have added additional information about the different treatment and management options in the manuscript discussion:

“Treatment of multifactorial disorders such as obesity is complex. In our approach, all patients received a multidisciplinary treatment advice tailored to their personal needs, including personalized dietary and physical activity advice (Fig 2). Furthermore, a monitoring and follow-up plan was developed for every patient. Local health care providers, including child health clinic physicians, general practitioners, general pediatricians, and psychologists, were contacted for local implementation of the care plan. In cases with severe hyperphagia, parental support by an educational therapist was offered to cope with the child’s behavior. Rehabilitation physicians were consulted when obesity interfered with performance of daily activities such as walking.[10] 

Establishing a main underlying cause of obesity can improve personalized treatment.[34] 

In all our 54 patients with an underlying medical cause, counseling about the diagnosis was given. This included advice pertaining to bariatric surgery, which has unclear long-term success rates for patients with underlying medical causes.[43, 48] Patients with genetic obesity were counseled by a clinical geneticist regarding inheritance, associated medical problems and reproductive decisions. Hormonal supplementation was started in case of hormonal deficiencies associated with specific genetic obesity disorders (such as growth hormone treatment in cases with leptin receptor deficiency).[49] In cases of syndromic obesity, the patients were evaluated for associated organ abnormalities or referred for disease-specific surveillance.[13, 25-32] In patients with cerebral/hypothalamic injury as cause of obesity and hyperphagia, dexamphetamine treatment was considered.[50] In patients with medication-induced obesity, evaluation of necessity and alternatives for the weight-inducing medication took place in collaboration with the prescribing physician. Follow-up studies are necessary to evaluate the different individual responses to these treatment options. Interesting novel developments are clinical trials with MC4R-agonists in patients with leptin-melanocortin pathway deficiencies, e.g. POMC and LEPR deficiency,[44] and glucagon-like peptide 1 (GLP-1) agonists for adolescents with obesity.[51] These GLP-1 agonists might also be a future treatment option for patients with genetic obesity disorders, as they have been shown to be equally as effective in adults with heterozygous MC4R mutations compared to adults without.[52] Recently, it was suggested that a subgroup of patients with severe early-onset obesity might have relative leptin deficiency and therefore might benefit from recombinant leptin administration.[53] However, the (long-term) effects of these new potential treatment options remain to be investigated.”

Reviewer #2: This is a very important study that utilized an extensive systematic diagnostic approach in a specialized obesity center to identify underlying causes of obesity among pediatric patients. This adds significantly to our knowledge base of the etiology of pediatric obesity and demonstrates that patients being referred to specialty clinics may have a higher prevalence of genetic obesity, medication induced obesity, and obesity due to cerebral injury.

Major comments:

1. More exploration as to why the investigators believe that BMI SDS was not significantly higher in patients with genetic obesity disorder when compared to patients without underlying medical causes. How does the obesogenic environment and socioeconomic factors in which we live play a role? This needs more emphasis in the discussion.

Authors’ response: We would like to thank the reviewer for their recognition of our manuscript. We added more information on the potential explanations of the comparable BMI SDS in the two mentioned patient groups in the discussion section: 

“A possible explanation is that severity of obesity increases the probability of being referred to a pediatric obesity center regardless of whether genetic obesity is diagnosed. Important factors that were more frequently present in the patients without underlying medical causes were psychosocial problems (DSM-5 diagnosis or involvement of authorities such as child protective services). These psychosocial problems might contribute to developing a higher BMI SDS.[47] On group level, we did not find evidence for significant differences in socio-economic status scores between patients with genetic obesity and patients without underlying medical causes, but individual differences in socio-economic factors and obesogenic environments might also play a role.” 

2. More information regarding how to specifically treat the different types of obesity would be beneficial in the discussion (Though I recognize it must be brief given word constraints).

Authors’ response: Based on this comment and the comment from reviewer #1, we have added additional information about the different treatment and management options in the manuscript discussion: 

“Treatment of multifactorial disorders such as obesity is complex. In our approach, all patients received a multidisciplinary treatment advice tailored to their personal needs, including personalized dietary and physical activity advice (Fig 2). Furthermore, a monitoring and follow-up plan was developed for every patient. Local health care providers, including child health clinic physicians, general practitioners, general pediatricians, and psychologists, were contacted for local implementation of the care plan. In cases with severe hyperphagia, parental support by an educational therapist was offered to cope with the child’s behavior. Rehabilitation physicians were consulted when obesity interfered with performance of daily activities such as walking.[10] 

Establishing a main underlying cause of obesity can improve personalized treatment.[34] 

In all our 54 patients with an underlying medical cause, counseling about the diagnosis was given. This included advice pertaining to bariatric surgery, which has unclear long-term success rates for patients with underlying medical causes.[43, 48] Patients with genetic obesity were counseled by a clinical geneticist regarding inheritance, associated medical problems and reproductive decisions. Hormonal supplementation was started in case of hormonal deficiencies associated with specific genetic obesity disorders (such as growth hormone treatment in cases with leptin receptor deficiency).[49] In cases of syndromic obesity, the patients were evaluated for associated organ abnormalities or referred for disease-specific surveillance.[13, 25-32] In patients with cerebral/hypothalamic injury as cause of obesity and hyperphagia, dexamphetamine treatment was considered.[50] In patients with medication-induced obesity, evaluation of necessity and alternatives for the weight-inducing medication took place in collaboration with the prescribing physician. Follow-up studies are necessary to evaluate the different individual responses to these treatment options. Interesting novel developments are clinical trials with MC4R-agonists in patients with leptin-melanocortin pathway deficiencies, e.g. POMC and LEPR deficiency,[44] and glucagon-like peptide 1 (GLP-1) agonists for adolescents with obesity.[51] These GLP-1 agonists might also be a future treatment option for patients with genetic obesity disorders, as they have been shown to be equally as effective in adults with heterozygous MC4R mutations compared to adults without.[52] Recently, it was suggested that a subgroup of patients with severe early-onset obesity might have relative leptin deficiency and therefore might benefit from recombinant leptin administration.[53] However, the (long-term) effects of these new potential treatment options remain to be investigated.”

Minor comments:

1. There are a couple of times throughout the manuscript that the authors write, “significantly more often”. This is awkward to read and should be revised for clarity.

Authors’ response: We thank the reviewer for the suggestion and have revised this phrasing throughout the manuscript.

2. The manuscript would benefit from editing by a native English speaker.

Authors’ response: We thank the reviewer for this suggestion and asked a native (American) English speaker to edit our manuscript. Please see the track changes version of our manuscript for further details.

---

## [Editor Report · Decision Letter 1]

27 Apr 2020

Identifying underlying medical causes of pediatric obesity: results of a systematic diagnostic approach in a pediatric obesity center

PONE-D-20-05457R1

Dear Dr. van den Akker,

We are pleased to inform you that your manuscript has been judged scientifically suitable for publication and will be formally accepted for publication once it complies with all outstanding technical requirements.

With kind regards,

David A. Buchner, PhD

Academic Editor

PLOS ONE
---

## [Editor Report · Acceptance letter]

29 Apr 2020

PONE-D-20-05457R1 

Identifying underlying medical causes of pediatric obesity: results of a systematic diagnostic approach in a pediatric obesity center 

Dear Dr. van den Akker:

I am pleased to inform you that your manuscript has been deemed suitable for publication in PLOS ONE. Congratulations! Your manuscript is now with our production department. 

With kind regards,

on behalf of

Dr. David A. Buchner 

Academic Editor

PLOS ONE